# Preventive training does not interfere with mRNA-encoding myosin and collagen expression during pulmonary arterial hypertension

**Thaoan Bruno Mariano[1], Anthony César de Souza Castilho[1], Ana Karenina Dias de Almeida Sabela[1], André Casanova de Oliveira[1], Sarah Santiloni Cury[2], Andreo Fernando Aguiar[3], Raisa de Jesus Dutra Dias[4], Antonio Carlos Cicogna[5], Katashi Okoshi[5], Luis Antonio Justulin Junior[2], Robson Francisco Carvalho[2], Francis Lopes Pacagnelli[1,4] ***

**1** Postgraduate Program in Animal Science, University of Western São Paulo (UNOESTE), Presidente Prudente, São Paulo, Brazil, **2** Department of Structural and Functional Biology, Institute of Biosciences, UNESP, Botucatu, São Paulo, Brazil, **3** Postgraduate Program in Physical Exercise in Health Promotion, Northern University of Paraná, Londrina, Paraná, Brazil, **4** Department of Physiotherapy, University of Western São Paulo (UNOESTE), Presidente Prudente, São Paulo, Brazil, **5** Department of Internal Medicine, Botucatu Medical School, UNESP, Botucatu, São Paulo, Brazil

* francispacagnelli@unoeste.br

**Data Availability Statement:** The additional data were generated in the indicated repository.

## Abstract

To gain insight on the impact of preventive exercise during pulmonary arterial hypertension (PAH), we evaluated the gene expression of myosins and gene-encoding proteins associated with the extracellular matrix remodeling of right hypertrophied ventricles. We used 32 male Wistar rats, separated in four groups: Sedentary Control (S, n = 8); Control with Training (T, n = 8); Sedentary with Pulmonary Arterial Hypertension (SPAH, n = 8); and Pulmonary Arterial Hypertension with Training (TPAH, n = 8). All rats underwent a two-week adaptation period; T and TPAH group rats then proceeded to an eight-week training period on a treadmill. At the beginning of the 11th week, S and T groups received an intraperitoneal injection of saline, and SPAH and TPAH groups received an injection of monocrotaline (60 mg/kg). Rats in the T and TPAH groups then continued with the training protocol until the 13th week. We assessed exercise capacity, echocardiography analysis, Fulton's index, cross-sectional areas of cardiomyocytes, collagen content and types, and fractal dimension (FD). Transcript abundance of myosins and extracellular matrix genes were estimated through reverse transcription-quantitative PCR (RT-qPCR). When compared to the SPAH group, the TPAH group showed increases in functional capacity and pulmonary artery acceleration time/pulmonary ejection time ratio and decreases in Fulton's index and cross-sectional areas of myocyte cells. However, preventive exercise did not induce alterations in *col1a1* and *myh7* gene expression. Our findings demonstrate that preventive exercise improved functional capacity, reduced cardiac hypertrophy, and attenuated PH development without interfering in mRNA-encoding myosin and collagen expression during PAH.

(https://zenodo.org/record/5233137/ DOI: 10.5281/zenodo.5233137).

**Funding:** This study was supported by São Paulo Research Foundation (FAPESP, grant #2016/11344-0, FLP). RDD was funded by a FAPESP fellowship #2018/12526-0. The authors thank Dr Dijon Henrique Salomé Campos for his help with animal care and Lauren Chrys Soato Marin Schaffer for technical assistance in Picrosirius red staining (FAPESP fellowship #2018/24317-7).

**Competing interests:** The authors have declared that no competing interests exist.

# Introduction

Pulmonary arterial hypertension (PAH) is a severe and disabling disease that causes right ventricular (RV) remodeling, compensatory hypertrophy, and RV heart failure (HF), the latter being the main prognostic determinant and common cause of death [1]. Alterations in myosin and extracellular matrix-related genes are possible mechanisms involved in the PAH heart failure phase. A study of the HF phase in isolated RV myocytes using monocrotaline demonstrated a reduction in ATPase activity in the myosin head [2]. These changes in myosin heavy chains have a critical role in HF. The chains are the main contractile proteins of the heart, and alterations can directly lead to decreased myocardial contractility [2, 3].

Other studies that used monocrotaline for inducing RV HF have shown cardiac collagen increases [4]. Cardiac collagen increases have been associated with different forms of overload pressure and increases to myosin with lower ATPase capacity [3]. RV failure is characterized by extensive fibrosis and changes to extracellular matrix protein expression, collagen, and metalloproteinases [5]. However, the gene expressions of myosins, collagen, and metalloproteinases have not been studied in the compensatory PAH hypertrophy phase [5].

Exercise is a common approach to control and limit cardiac damage. It promotes changes in cardiac remodeling and brings benefits in human and animal models with RV hypertrophy [6, 7]. Preventive exercise promotes a cardioprotective effect in PAH, as it improves RV function and softens the evolution of the pathological cardiac remodeling process [8, 9]. Various molecular mechanisms have been studied to evaluate cardiac functional improvement from preventive training. These mechanisms include the expression of calcium transit genes, regulation of TNF superfamily cytokines, and the quantification of myosin isoforms. However, the effects of changes to the extracellular matrix gene expression and myosins on PAH with compensated RV hypertrophy have not been explored. In the HF phase, pathological remodeling is impossible to reverse by therapy. Thus, approaches to treat compensatory hypertrophy are important to alleviate dysfunctional impairment [10].

PAH with compensated RV hypertrophy often evolves to HF and results in high death rates and frequent hospitalizations. This validates the necessity of elucidating effects of preventive training and molecular mechanisms on RV hypertrophy, as this phase precedes HF. Our study hypothesizes that preventive aerobic training mitigates the gene changes in the compensated ventricular hypertrophy phase in monocrotaline-induced PAH rats. We investigate the influence of preventive aerobic training in rats with compensated RV hypertrophy on the gene expression of myosin heavy chains and the extracellular matrix.

# Materials and methods

## Ethical approval

The experimental protocols used in this study were approved by the Animal Experimentation Ethics Committee (CEUA) from the University of Western São Paulo, Presidente Prudente, São Paulo, Brazil (protocol numbers 2483 and 2484). The rats received care in accordance with the "Laboratory Animal Care Principles" formulated by the National Society for Medical Research and the "Guide for the Care and Use of Laboratory Animals" from the Laboratory Animal Research Institute [11].

## Experimental groups

We used 32 male Wistar rats. The rats were 2 months of age and had an average weight of 205 ±17.43 g. We received the rats from the Central Animal Facility of the University of Western São Paulo, UNOESTE. They were housed in a temperature-controlled room (23°C) with a

relative humidity of 50–60%. They were exposed to an inverted 12 h–12 h light–dark cycle. Food and water were supplied *ad libitum*.

The rats were randomly distributed into four experimental groups that had eight rats per group: Sedentary Control (S, n = 8); Control with Training (T, n = 8); Sedentary with Pulmonary Arterial Hypertension (SPAH, n = 8); and Pulmonary Arterial Hypertension with Training (TPAH, n = 8).

## Experimental design

The rats in the T and TPAH groups were submitted to an aerobic training protocol on a treadmill for 13 weeks. The first two weeks were periods of adaptation for all groups because the rats were conducted an incremental exercise test to assess their exercise capacity. At the beginning of the 11th week, S and T groups received an intraperitoneal injection of saline, and SPAH and TPAH groups received an injection of monocrotaline. Twenty-four hours after the injection, T and TPAH groups continued with the aerobic training protocol for another 3 weeks [12]. For load adjustment and training continuity, we analyzed the lactate threshold of the rats [13]. Echocardiographic evaluations and analysis of HF signs/weight were performed 48 hours after their final exercise session. The rats were then euthanized. The heart was removed and dissected, and the atria (AT), RV, and left ventricle (LV) + intraventricular septum (IVS) were weighed. We then performed histology, fractal dimension (FD), and gene expression of myosin, collagen, and metalloproteinase. Details of the experimental design (Fig 1) are described below.

## Preventive training

Rats from the T and TPAH groups were submitted to an adapted treadmill aerobic training protocol (model TK 1, IMBRAMED). The protocol consisted of 13 total weeks, five days a week. The first 2 weeks were for adaptation (familiarization). After, the rats performed the exercises for 8 weeks with gradual increases in intensity, as previously described. The rats were then injected with monocrotaline or saline and performed the exercises for 3 more weeks, in the dark moment. [12, 14].

## Incremental exercise test

To adjust the exercise speed and assess functional capacity, the rats were submitted to incremental stress tests 24 hours after monocrotaline administration [12, 15]. All exercise was performed with 0% slope. The tests began with a warm-up at 0.5 km/h, followed by five minutes of rest. The speed was then increased by 0.2 km/h every 3 minutes until lactate reached a 1 mmol/l comparative value or exhaustion [13]. Exhaustion was defined as the moment when rats could not continue running for 3 consecutive minutes. After each increased load, the rats were manually removed from the exercise area for 1 minute for blood collection. Blood samples were taken from rat tails every 3 minutes. We used an Accutrend Plus lactometer (Roche, Barcelona, Spain). The device was calibrated to the manufacturer's specifications. The calculation for stipulating maximum velocity was performed using the arithmetic mean of all experimental group velocities until lactate threshold or exhaustion [16]. Lactate threshold was defined as the rate of rotation without a lactate increase of 1.0 mmol/l above the blood-lactate concentration [14, 17]. We used an adapted version of the protocol created by Carvalho *et al.* [13]. Twenty-four hours after monocrotaline administration and the training periods, we evaluated delta change (Δ) of the lactate threshold velocity 24 hours after monocrotaline application and after training.

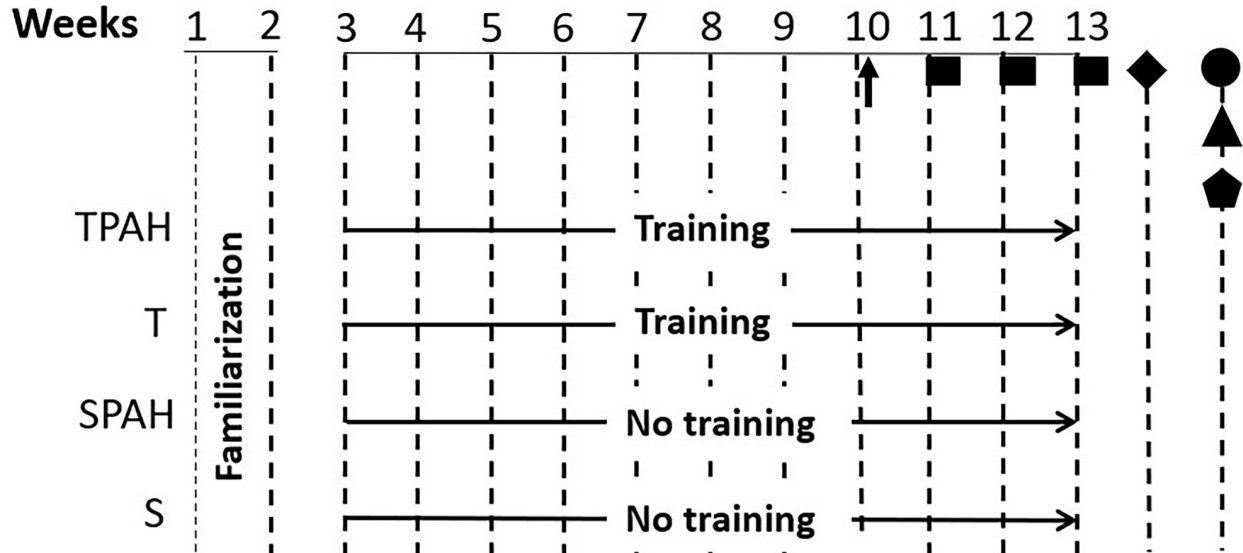

**Fig 1. Experimental design.** S (n = 8): Sedentary Control; SPAH (n = 8): Sedentary Pulmonary Arterial Hypertension; T (n = 8): Control with Training; TPAH (n = 8): Pulmonary Arterial Hypertension with Training.

### Induction of pulmonary hypertension

At the end of the 10th week, a single saline dose (NaCl 0.9%) was administered intraperitoneally to the S and T groups to ensure that all rats were subjected to the same degree of stress. The protocol for induction of PAH and RV hypertrophy was performed in the rats of the SPAH and TPAH groups with an intraperitoneal injection of a single 60 mg/kg monocrotaline dose (PHL89251; Sigma Chemical, St Louis, United States [18]. Monocrotaline is a pyrrolizidine alkaloid that induces pulmonary vascular disease with RV hypertrophy (21 days) and HF (28–30 days) [19, 20]. The monocrotaline-induced PH experimental model with a dose of 60 mg/kg effectively leads to rapid and progressive RV hypertrophy (21 days), thus allowing an enhanced sensitivity of detection of right dysfunction due to a large magnitude of change in a short time frame [21, 22].

### Echocardiographic evaluation

Two days after the end of the training protocol, we performed echocardiographic evaluations using an echocardiogram (General Electric Medical Systems, Vivid S6, Tirat Carme, Israel)

equipped with a 5–11.5 MHz multifrequency probe. The rats were intraperitoneally anesthetized with ketamine (50 mg/kg$^{-1}$; Dopalen$^{®}$) and xylazine (0.5 mg/kg$^{-1}$; Anasedan$^{®}$).

The following LV variables were measured: diastolic (LVDD) and systolic (LVSD) diameters, ratio of E and A waves (E/A), percentage of endocardial shortening (EFS), isovolumetric relaxation time (IVRT), heart rate frequency (HR), ejection fraction (EF), and posterior wall shortening velocity (EPVP). The indirect measures of RV afterload (pulmonary artery acceleration time [PAAT] and pulmonary ejection time [PET]) were also measured. To account for heart rate variability, PAAT was adjusted to PET and presented as PAAT/PET. PAAT and PET are noninvasive measures of RV afterload that provide accurate estimates of invasive pulmonary vascular resistance, pulmonary arterial pressure, and pulmonary arterial compliance in children with PH and PH mouse and rat models. [14, 23–26].

## Euthanasia

After the echocardiographic evaluation (48 hours), the rats were weighed and then euthanized with an intraperitoneal dose of sodium pentobarbital (150 mg/kg), in accordance with CON-CEA-Brazil recommendations [27]. During the euthanasia process, two observers determined the presence/absence of clinical and pathological congestive HF features. The clinical findings suggestive of HF were tachypnea and labored respiration. Pathologic assessment of HF included subjective evaluation of pleuropericardial effusion, atrial thrombi, ascites, and liver congestion.

## Evaluation of anatomical parameters

We evaluated final body weight, as excessive weight loss can indicate cardiac cachexia [20, 21]. Hearts were removed, dissected, and separated at the AT, RV, and LV + intraventricular septum (IVS). ATs, RVs, and LVs + IVSs were then weighed. For Fulton's index of RV hypertrophy, the ratio of RV weight to LV + IVS was calculated [24, 28]. The lungs and liver were also removed, weighed, and stored in an oven for 48 hours. The rats were then weighed again to calculate the wet/dry weight ratio, which was used to evaluate signs of HF [13].

## Histology and fractal analysis

The RV was divided into two parts. One part was fixed in 10% buffered formalin solution for 48 hours, and the other was used for gene expression analysis. After fixation, the tissues were placed on paraffin blocks. The histological sections were stained on slides with haematoxylin–eosin solution (HE) to measure the cross-sectional areas of the cardiomyocytes, using a LEICA microscope (model DM750, Leica Microsystems, Wetzlar, Germany). At least 50 cardiomyocyte cross-sectional areas were measured from each RV. After choosing sites with the most cells on a cross-section, different fields were captured and analyzed. The selected myocytes were cross-sectioned, had a round shape and visible central nucleus, and were located in the subendocardial layer of the RV muscular wall. All images were captured by video camera at 40X objective with 400x magnification [12].

Histological sections of the RV myocardial interstitium were stained on histological slides by the picrosirius technique for collagen visualization. The cardiac tissue images were captured by a computer coupled to a camcorder. Digital images from the LEICA DM LS microscope (model DM750, Leica Microsystems, Wetzlar, Germany) were sent to a computer equipped with Image-Pro Plus (Media Cybernetics, Silver Spring, United States). The red collagen color (picrosirius) was turned blue to reveal the percentage of collagen in relation to the total area of the image. Twenty fields of each RV were analyzed using a 40X objective with 400x magnification. The chosen fields were far from the perivascular region [29]. Picrosirius staining viewed

under polarized light enables the differentiation of type I (red) and type III (green) collagen. We used Image J software to measure the medium of coloration of these collagens in relation to the total image area [30]. Binarized photographs and the box-counting method using ImageJ software were used for FD analysis. The software used box-counting with two dimensions. This allowed for the quantification of pixel distribution without interference from the texture of the image. This results in two images (binarized and gray level) with the same FD. The analysis of the fractal histological slides was based on the relation between the resolution and the evaluated scale. The result was quantitatively expressed as the FD of the object with DF ¼ (Log Nr/Log r_1; Nr as the quantity of equal elements needed to fill the original object with scale applied to the object). FD was calculated using the ImageJ software set between 0 and 2, without distinguishing different textures [31–33].

## Real-time polymerase chain reaction after reverse transcription (RT-qPCR)

Total RNA was extracted from RV tissue using TRIzol (ThermoScientific, Waltham, United States) and then treated with DNAse deoxyribonuclease I (ThermoScientific) following the manufacturer's instructions. RNA integrity was evaluated by agarose gel electrophoresis for visualization of ribosomal RNAs. The High Capacity Reverse Transcriptional Kit (Thermo-Scientific) was used for the synthesis of complementary RNA (cDNA) from 1000 ng of total RNA for each sample. Aliquots of cDNA were then submitted to real-time PCR reaction using a customized assay containing the following Taqman (Applied Biosystems, Foster City, United States) probes specific to each gene: Rattus norvegicus myosin heavy chain 6 (*myh6*, Rn01489272_g1) mRNA, Rattus norvegicus myosin heavy chain 7 (*myh7*, Rn01488777_g1), Rattus norvegicus myosin heavy chain 7B (*myh7b*, Rn01536269_m1), Rattus norvegicus collagen type I alpha 1 chain (*col1a1*, Rn01463848_m1) mRNA, Rattus norvegicus collagen type I alpha 2 chain (*col1a2*, Rn00756233_m1), Rattus norvegicus collagen type 3 alpha 1 chain (*col3a1*, Rn01437681_m1) mRNA, and Rattus norvegicus metalloproteinases 2 (*mmp2*, Rn01538170_m1) mRNA. The Taqman™ Universal Master Mix II (AppliedBiosystems) and the StepOne Plus system (ThermoScientific) were used for qPCR. All samples were analyzed in duplicates. The cycling conditions were at 50 ˚C for 2 minutes and 95 ˚C for 10 minutes. This was followed by 40 cycles of denaturation at 95 ˚C for 15 seconds and the final extension at 60 ˚C for 1 minute. Gene expression was quantified relative to the values of the S group after normalization by expression levels of the beta-actin reference gene (*Actb*, Rn00667869_m1) using the $2^{-\Delta\Delta Ct}$ method [34].

## Data analysis

Statistical analyses were performed using GraphPad Prism software (Graph-Pad Software, La Jolla, United States). The Shapiro-Wilk test was used to assess data normality. One-way ANOVA and Tukey's post test were used to analyze the anatomical parameters, echocardiogram data, collagen fiber type. The Kruskal-Wallis test and Dunn's post test were used to analyze data from the collagen interstitial fraction, FD, and gene expression. Data were expressed in mean ± standard deviation, or median, minimum, maximum. The significance level was considered when $p < 0.05$.

## Results

### Exercise capacity

When compared to the S, T, and TPAH groups, we observed in the SPAH group a decrease in the delta change (Δ) of the lactate threshold velocity 24 hours after monocrotaline application

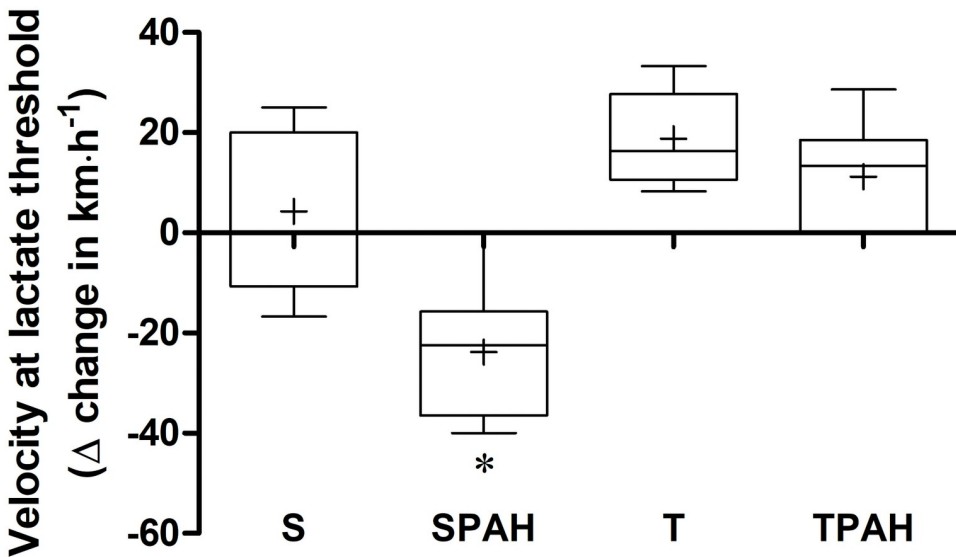

**Fig 2. Delta change in the velocity at lactate threshold from after 24 hours of monocrotaline application to post-training in the sham (S, n = 8), pulmonary arterial hypertension (SPAH, n = 8), trained (T, n = 8), and trained with pulmonary arterial hypertension (TPAH, n = 8) groups.** The box plot shows the median (line) and mean (+), interquartile range (box), and the maximum and minimum values (whiskers). * p < 0.001 compared to S, T, and TPAH. One-way ANOVA and Tukey's post hoc.

and after training. Thus, aerobic exercise improved functional capacity in the TPAH group (Fig 2).

### Echocardiographic evaluation

The LV echocardiographic evaluation is presented in Table 1. LVDD was lower in the SPAH group when compared to rats in the control group. LVDD was higher in the T group when compared to rats in the control group. We observed decreased PAAT/PET in the SPAH group when compared to the S group. An improvement of PAAT/PET in the TPAH group was also recorded (Fig 3).

**Table 1. Left ventricle echocardiographic evaluation.**

| PARAMETERS | S (n = 8) | SPAH (n = 8) | T (n = 8) | TPAH (n = 8) |
|---|---|---|---|---|
| HR (bpm) | 310.71±46.58 | 331.66±35.65 | 314±38.25 | 325.44±42.81 |
| LVDD (mm) | 7.84±0.54 | 7.02±0.73* | 8.25± 0.50* | 7.37±0.94 |
| LVSD (mm) | 4.14±0.48 | 3.59±0.47 | 4.39± 0.39 | 3.45±0.81 |
| EFS | 47.33±3.56 | 48.72± 6.05 | 46.86± 2.28 | 53.50± 7.07 |
| E/A | 1.55± 0.31 | 1.07±0.36 | 1.37± 0.19 | 1.12± 0.37 |
| PWSV (mm/s) | 34.21±0.90 | 37.37± 3.43 | 37.85± 3.30 | 37.38±3.85 |
| IVRT (ms) | 22.57± 3.59 | 31.55± 9.40 | 22.01± 3.57 | 28.66±7.26 |
| Ejection Fraction | 0.85±0.03 | 0.86±0.04 | 0.84±0.01 | 0.89±0.04 |

Data are expressed as mean ± standard deviation. Heart rate (HR), diastolic (LVDD) and systolic (LVSD) diameters, percentage of endocardial shortening (EFS), the ratio of E and A waves (E/A), posterior wall shortening velocity (PWSV), time isovolumetric relaxation rate (IVRT), ejection fraction (EF). S (n = 8): Sedentary Control; SPAH (n = 8): Sedentary Pulmonary Arterial Hypertension; T (n = 8): Control with Training; TPAH (n = 8): Pulmonary Arterial Hypertension with Training.

* p<0.05 compared to S.

One-way ANOVA and Tukey's post test.

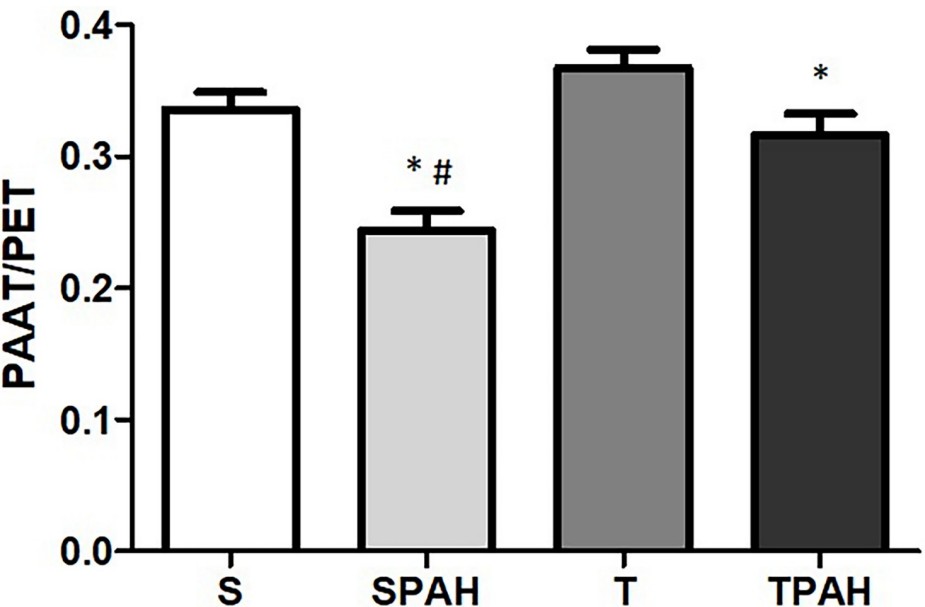

**Fig 3. RV echocardiographic evaluation; PAAT/PET-PAAT: Pulmonary artery acceleration time, PET: Pulmonary artery ejection time; S (n = 8): Sedentary Control; SPAH (n = 8): Sedentary Pulmonary Arterial Hypertension; T (n = 8): Control with Training; TPAH (n = 8): Pulmonary Arterial Hypertension with Training.** * p<0.05 compared to S; # compared to TPAH. One-way ANOVA and Tukey's post hoc.

## Group characterization and anatomic parameters

There was no excessive weight loss in PAH groups (S = 406±28.77g; SPAH = 387±66.47g; T = 433±35.14g; TPAH = 391±27.41g, p<0.05). We did not observe tachypnea/labored respiration, pleural effusion, or liver congestion evidence from HF in PAH groups. All rats with PAH (n = 16) presented atria hypertrophy (S = 0.22 ± 0.03 g; SPAH = 0.37 ± 0.18 g; TPAH = 0.36 ± 0.19 g, p< 0.05) and significantly increased Fulton's index. Preventive exercise decreased Fulton's index, indicating a reduction of RV hypertrophy in TPAH group (S = 0.24 ±0.03; SPAH = 0.46±0.12; TPAH = 0.33±0.93; p<0.05).

## Histological and fractal analysis

Fiber cross sectional areas were higher in SPAH groups. Physical exercise attenuated hypertrophy in TPAH animals by decreasing the sectional area of the cardiomyocytes (S = 62.9 ± 6.37 $\mu m^2$; SPAH: 104.88 ± 21.83 $\mu m^2$; TPAH = 89.23 ± 7.99 $\mu m^2$, p<0.05) (Fig 4). There was no increase in the percentage of interstitial collagen in the PAH groups (p> 0.05) (Fig 5). The images under polarized light show a greater presence of red collagen (type I) compared to green (type III) in all groups (Fig 5). Type I collagen increased in the SPAH and TPAH groups when compared to the S group (S = 22.53 ± 0.9 ua; SPAH = 25.5 ± 1.5 ua; TPAH = 25.92 ± 1.18 ua, p < 0.05). Type III did not show statistical difference (p>0.05). There were no changes in FDs between the analyzed groups (p> 0.05) (Fig 4).

## Relative gene expression

When compared to the control group, we observed in the SPAH group that *Myh7* gene expression had a tendency to increase (S vs SPAH, p = 0.0939). Comparisons with the control group show that preventive exercise increased *Myh7* gene expression in the TPAH group (S vs. TPAH, p = 0.0242). *Col1a1* expression was higher in the SPAH and TPAH groups when

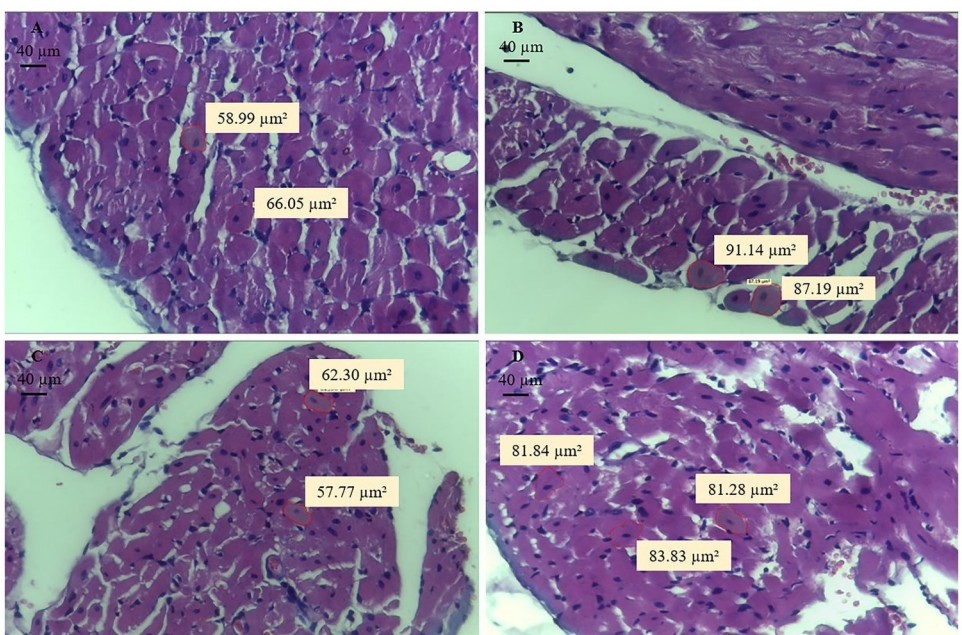

**Fig 4. Area of cardiomyocytes in the epicardial region stained in haematoxylin–eosin, 40x objective and 400x magnification.** A. S (n = 8): Sedentary Control; B. SPAH (n = 8): Sedentary Pulmonary Arterial Hypertension; C. T (n = 8): Control with Training; D. TPAH (n = 8): Pulmonary Arterial Hypertension with Training.

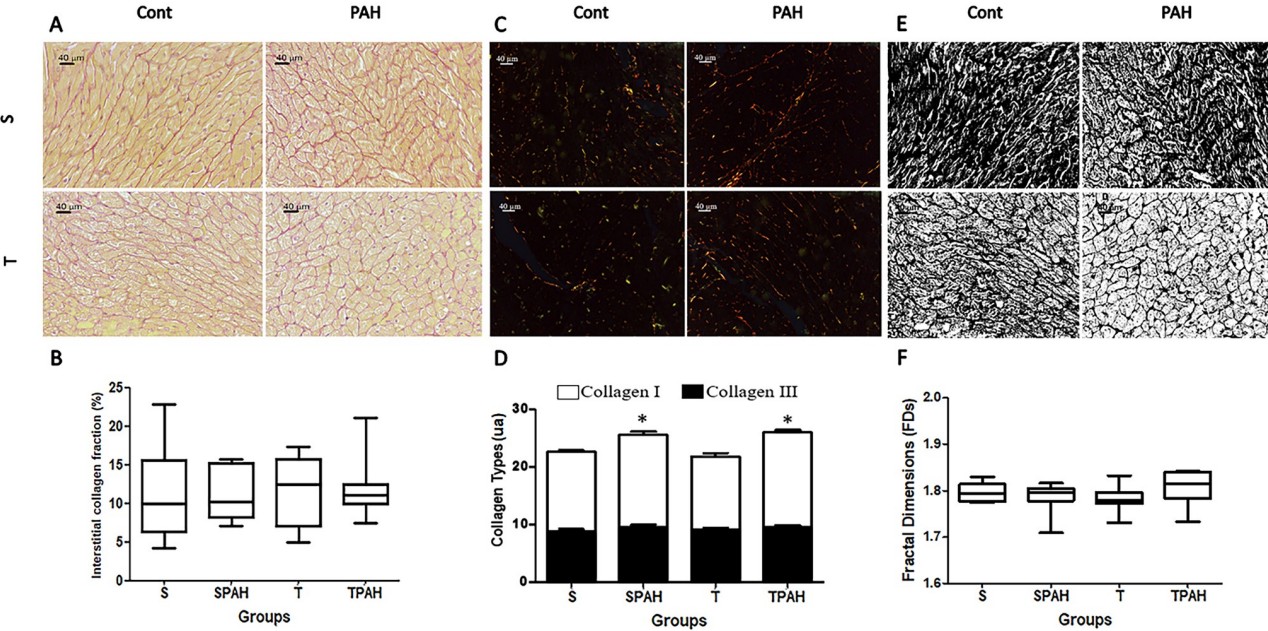

**Fig 5.** A: Cross-sections of the cardiac muscle were stained by the picrosirius red technique (PSR) and viewed with 40x objective and 400x magnification. B. Quantitative analysis of PSR-stained sections. C. PSR observed under polarized light with 40x objective and 400x magnification. The red and green colors are the collagens I and III, respectively. D. Quantitative analysis of collagen types. E. Histological sections of the cardiac muscle stained with Picrosirius-red staining after binarization at 40x objective and 400 magnification. F. Quantitative analysis of fractal dimension. Cont: controls groups. PAH: Pulmonary Arterial Hypertension groups; S: sedentary; T: training. S (n = 8): Sedentary Control; T (n = 8): Control with Training; SPAH (8): Sedentary Pulmonary Arterial Hypertension; TPAH (n = 8): Pulmonary Arterial Hypertension with Training. * $p < 0.05$ compared to S One-way ANOVA and Tukey's post hoc or Kruskal-Wallis and Dunn's post hoc.

compared to the S and T groups (S vs. SPAH, S vs. TPAH, T vs. TPAH, p = 0.0008); the other genes did not present statistically significant differences (Fig 6).

## Discussion

The aim of this study was to evaluate the effects of exercise on the molecular mechanisms of gene expression of myosins and the extracellular matrix compensated ventricular hypertrophy phase in monocrotaline-induced PAH rats. Our findings showed that *Col1a1* and *Myh7* genes were altered in the hypertrophy phase in PAH. Despite that preventive training resulted in greater functional capacity, less cardiac hypertrophy, and attenuation of PAH progression, exercise did not modify any of the analyzed genes.

Physical inactivity is considered a risk factor for cardiovascular diseases. Studies indicate that regular, moderate exercise (e.g., walking) in healthy individuals is associated with a reduction in the incidence and evolution of cardiovascular events [35–38]. Studies have shown that individuals with regular fitness routines before PAH development receive cardioprotective benefits that mitigate PAH-related alterations in the heart [8, 12].

When compared to sedentary rats, the findings from our study showed better functional capacity in rats that performed preventive training for 8 weeks before a monocrotaline injection and then continued with the training for 3 weeks after the injection. When examining the velocity variation in the lactate threshold in the first and last tests, we observed that the rats in the SPAH group had less functional capacity than the other groups. We also observed TPAH group rats had functional capacity that was similar to healthy rats. These functional capacity improvements are likely caused by increased $VO_2max$, with peripheral muscle improvements and enhanced cardiac function [39–41]. Various studies have reported increased exercise tolerance in humans and animals with PAH that had an exercise regimen [4, 42–44]. Decreased functional capacity is an indication of poor prognosis for PAH, and preventive physical exercise has been shown to have a fundamental role in improving the condition.

In our study, an echocardiographic evaluation demonstrated decreased PAAT/PET in SPAH group. PAAT and PET are noninvasive measures of RV afterload; they provide accurate estimates of invasive pulmonary vascular resistance, pulmonary arterial pressure, and pulmonary arterial compliance in children with PH and in PH mouse and rat models [23–25]. Low PAAT is also associated with differing degrees of RV dysfunction in PH patients, which parallels the relationship of PAAT with pulmonary vascular resistance and pulmonary arterial compliance [25]. The effects of low PAAT imply that RV afterload is a major contributor to the observed differences in RV function. The development of significant decreases to RV function may lead to a slower rise in ejection velocities early in systole and increased PAAT. We showed increased PAAT/PET in the TPAH group, which implies that preventive exercise reduced the RV pressure overload in PAH. Such beneficial effects of exercise in RV function have been demonstrated in various other studies [6, 8, 12]. However, previous studies have not reported increases in PAAT or PAAT/PET from a treadmill training protocol [4, 45]. The divergent findings in the literature show that installation of severe PAH by monocrotaline injection counters all potentially beneficial effects of aerobic exercise training; this highlights the importance of preventive exercise in cases of hereditary pulmonary hypertension and early stages of PAH to prevent its worsening.

Our findings also demonstrated an increase in LVDD. However, we did not observe an increase in the ejection fraction. Physical training increases venous return, which is characterized by an increase in LV afterload and consequent cardiac remodeling with an LVDD increase [46]. The enhanced cardiac contractile function promoted by exercise training is also related to the modest increase in LV muscle mass and LV adaptive cardiac hypertrophy, which

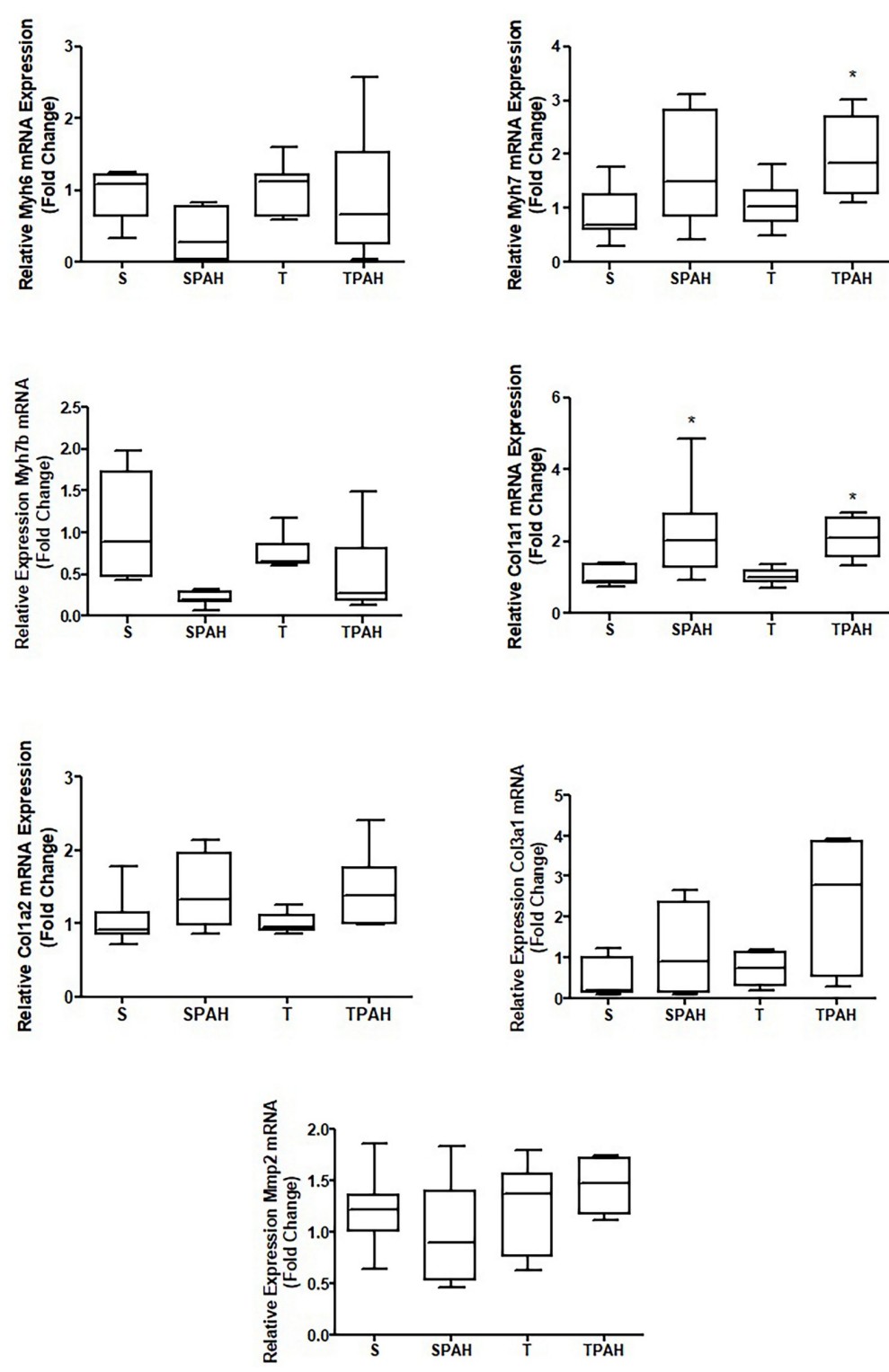

**Fig 6.** The mRNA abundance of A. *Myh6* gene expression. B. *Myh7* gene expression. C. *Myh7b* gene expression. D. *Col1a1* gene expression. E. *Col1a2* gene expression. F. *Col3a1* gene expression. G. *Mmp2* gene expression in RV of experimental groups. S (n = 8): Sedentary Control; T (n = 8): Control with Training; SPAH (8): Sedentary Pulmonary Arterial Hypertension; TPAH (n = 8): Pulmonary Arterial Hypertension with Training. p <0.05 compared to S. Kruskal-Wallis and Dunn's post hoc.

is paralleled by angiogenesis, changes in fibrillar collagen content, and organization that enhances cardiac pump function [47]. The insignificant changes to the ejection fraction in our study may be related to the exercise intensity or an insufficient time period for significant LV improvements.

Monocrotaline also promoted RV hypertrophy in SPAH and TPAH groups; exercise had a beneficial effect on this parameter. The benefits of exercise in reducing RV afterload and on ventricular hypertrophy have been reported in previous studies [4, 8, 12]. In a study that exposed male rats to a 4-week exercise regimen (treadmill) or sedentary period prior to the administration of monocrotaline (60 mg/kg), Nogueira-Ferreira et al. [8] showed that exercise preconditioning prevented cardiac hypertrophy by decreasing RV cardiomyocyte cross-sectional areas. In another study, male Wistar rats received monocrotaline injections (60 mg/kg) to induce PAH and were divided into three groups: normal cage activity, exercise training in early PAH stage, and exercise training in late PAH stage. The study showed that exercise in early and late PAH stages reverted RV/Boby weight and FI. However, one study reported that 4 weeks of moderate aerobic exercise did not affect RV hypertrophy in rats with monocrotaline-induced PAH [44]. Rats in studies by Colombo et al. [6, 48, 49] and Zimmer et al. [45] performed moderate intensity exercise on a treadmill for three weeks; when compared to controls, no changes in RV hypertrophy was observed. The severity of PAH, intensity, type, and duration of training may explain the lack of changes.

RV adaptations in PAH are influenced by the activity of myosins and the extracellular matrix. The shift from alpha- to beta-myosin heavy chain (MHC) is widely used as an indicator of cardiac maladaptive remodeling [50]. In the ventricular hypertrophy phase without HF, we detected that m*yh7* gene expression (gene responsible for the increase in β-MHC) tended to increase; preventive exercise did not have an effect on this change. However, this tendency may have been influenced by the dispersion of the expression levels of this gene. This possible adaptation in *myh7* gene expression may reflect a more economical energetic phenotype, as beta-MHC can generate a cross-bridge force that provides more economical energy consumption in the ventricular hypertrophy phase [50]). In more advanced stages of PAH, the shift to the slower beta-MHC isoform was observed in sedentary PAH animals and associated with reduced myosin ATPase enzyme velocity. This reduction slows the myocyte contraction rate [8, 51, 52]. Another study demonstrated that high levels of heart alpha-MHC isoform were expressed 4 weeks after monocrotaline injection, which could explain improvements to cardiac output as a cardiac function that is linearly related to MHC content [4, 53]. Contrary to our findings, Nogueira et al. [8] showed that exercise preconditioning may have contributed to the improved cardiac function in monocrotaline-treated animals by promoting reductions to the beta/alpha-MHC ratio. We interpret these different outcomes mainly as consequences of dissimilarities in the intensity, duration, and type of training.

In addition to myosins, the extracellular matrix supports and connects all structures and assists cardiac function [7, 54–58]. To our knowledge, our study is the first that investigates the effects of exercise and compensated PAH training on gene-encoding proteins associated with different collagen types and their organization (FD). In our study, an increase of *col1a1* gene expression and collagen type 1 fibers were observed in the PAH cardiac hypertrophy phase in PAH; this effect was not dependent on exercise. Although we detected gene expression and collagen type changes, we did not observe fibrosis; this can be explained by PAH severity. The rats in our study did not have signs of HF, unlike other studies in the literature [4, 6, 59]. Our results are in agreement with a study by Nadadur et al. [4] that showed upregulation of gene expression of collagen I in PH using monocrotaline (60 mg/kg). Collagen types I and III both markedly increased cardiac pressure-overload due to the mechanical stress-activating fibroblasts and NF-κB pathway activation [8, 60]. Although the expression profile of fibrillar

collagens in the early stage exhibits the prevalence of collagen III, type I collagen displays more intense and prolonged upregulation in both compensatory and decompensatory stages and eventually leads to decreased myocardial distensibility [61–63]. Despite alterations in gene expression and type I collagen, fractal evaluation showed that tissue collagen organization was preserved. FD is an effective method to evaluate cardiac morphological changes induced by ventricular dysfunction [33, 64]. Using HE staining, Pacagnelli et al. [64] reported increased FD of morphological and nuclear aspects of cardiomyocytes; these increases were associated with cardiac hypertrophy and dysfunction. The results in our present study were likely affected by the evaluation type, as we analyzed the organization of the extracellular matrix. The matrix metalloproteinase MMP2 did not change in PAH groups in our study. MMPs 2 and 9 damage cardiomyocytes when increased, but these changes in metalloproteinase only occur with the development of HF [65].

We did not observe that preventive training decreased type I collagen gene expression, although we noted that exercise attenuated PAAT/PET and cardiac hypertrophy. Factors independent of cardiac hypertrophy may be related to collagen, such as the action of miRNA. [65–70]. However, various studies have reported that combined exercise modalities can change collagen types in pathological conditions (e.g., infarction) and may be associated with hypertrophy reversal [71, 72]. Similar to myosins, collagen can be affected by exercise modality, frequency, duration, intensity, and the disease phase when training starts [8]. Future studies should investigate the effects of other mechanisms, pathways, and variations in intensity/type of exercise on the compensated hypertrophy phase of PAH.

The advantages of MCT models are technical simplicity, reproducibility, and relatively low cost. However, because the animals present heart failure at 4 weeks, it is a model that does not allow the evaluation of the effects of longer period exercise protocols [1, 4, 6, 8, 9].

Contrary to our hypothesis, our findings showed that only *Col1a1* and *Myh7* genes were altered in the compensated hypertrophy stage, and preventive training did not modify these genes. However, preventive training resulted in greater functional capacity, less cardiac hypertrophy, and attenuation of PAH progression even without interfering in mRNA-encoding myosin and collagen expression during PAH.

## Acknowledgments

We would like to thank Eric Schloeffel for his help with English editing.

## Author Contributions

**Conceptualization:** Anthony César de Souza Castilho, Ana Karenina Dias de Almeida Sabela, Andreo Fernando Aguiar, Raisa de Jesus Dutra Dias, Francis Lopes Pacagnelli.

**Data curation:** Thaoan Bruno Mariano, Ana Karenina Dias de Almeida Sabela, André Casanova de Oliveira, Andreo Fernando Aguiar, Robson Francisco Carvalho, Francis Lopes Pacagnelli.

**Formal analysis:** Thaoan Bruno Mariano, Anthony César de Souza Castilho, Ana Karenina Dias de Almeida Sabela, Sarah Santiloni Cury, Andreo Fernando Aguiar, Katashi Okoshi, Robson Francisco Carvalho, Francis Lopes Pacagnelli.

**Funding acquisition:** Raisa de Jesus Dutra Dias, Francis Lopes Pacagnelli.

**Investigation:** Thaoan Bruno Mariano, Anthony César de Souza Castilho, André Casanova de Oliveira, Luis Antonio Justulin Junior, Francis Lopes Pacagnelli.

**Methodology:** Ana Karenina Dias de Almeida Sabela, André Casanova de Oliveira, Sarah Santiloni Cury, Andreo Fernando Aguiar, Raisa de Jesus Dutra Dias, Katashi Okoshi, Luis Antonio Justulin Junior, Francis Lopes Pacagnelli.

**Project administration:** Anthony César de Souza Castilho, Antonio Carlos Cicogna, Francis Lopes Pacagnelli.

**Supervision:** Antonio Carlos Cicogna, Francis Lopes Pacagnelli.

**Validation:** Francis Lopes Pacagnelli.

**Writing – original draft:** Thaoan Bruno Mariano, Anthony César de Souza Castilho, Francis Lopes Pacagnelli.

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
