## [Decision Letter · Decision Letter 0]

2 Feb 2021

PONE-D-20-38767

Preventive training interferes with mRNA-encoding myosin 7 and collagen I expression during pulmonary arterial hypertension

PLOS ONE

Dear Dr. Pacagnelli,

Thank you for submitting your manuscript to PLOS ONE. After careful consideration, we feel that it has merit but does not fully meet PLOS ONE’s publication criteria as it currently stands. Therefore, we invite you to submit a revised version of the manuscript that addresses the points raised during the review process.

We look forward to receiving your revised manuscript.

Kind regards,

Michael Bader

Academic Editor

PLOS ONE

Journal Requirements:

2. We noticed you have some minor occurrence of overlapping text with the following previous publications, which needs to be addressed:

- http://bdtd.unoeste.br:8080/jspui/bitstream/jspui/995/2/Thaoan.pdf

- https://journals.physiology.org/doi/full/10.1152/japplphysiol.01349.2011

- journals.physiology.org/doi/full/10.1152/japplphysiol.01349.2011

In your revision ensure you cite all your sources (including your own works), and quote or rephrase any duplicated text outside the methods section.

Further consideration is dependent on these concerns being addressed.

Reviewers' comments:

Reviewer's Responses to Questions

**Comments to the Author**

1. Is the manuscript technically sound, and do the data support the conclusions?

Reviewer #1: No

Reviewer #2: No

2. Has the statistical analysis been performed appropriately and rigorously? 

Reviewer #1: I Don't Know

Reviewer #2: Yes

3. Have the authors made all data underlying the findings in their manuscript fully available?

Reviewer #1: No

Reviewer #2: Yes

4. Is the manuscript presented in an intelligible fashion and written in standard English?

Reviewer #1: No

Reviewer #2: Yes

5. Review Comments to the Author

Reviewer #1: Mariano et al performed a study in vivo to investigate if aerobic training can mitigate (preventive effect) the gene changes in the compensated ventricular hypertrophy phase in monocrotaline-induced PAH rats. The study proposal is relevant; however, several issues must be clarified in order to improve the manuscript. Following the points that must be reviewed.int

Major issues:

1) The experimental design is confusing. A topic is needed explaining the model's time course with an illustration.

2) There is a lack of information about the monocrotaline treatment. 2.1) Why was used the dose of 60 mg/kg? 2.2) Were three weeks of treatment sufficient to induce cardiac disfunction? 2.3) How was monocrotaline administered? 2.4) What was monocrotaline come from? Which company, code and lot? 2.4) Was there an adverse effect such as excessive weight loss? 2.5) What is the limitation of using this PAH model? Important references about this PAH model were lacking.

3) Physical training was performed before and during the development of PAH. Did physical training interfere with the effect of MCT in inducing PAH or did it actually prevent the development of PAH once the condition is established? Is there any data that shows that physical training does not interfere with the action of MCT?

4) What phase of the 12-hour light / dark cycle did the physical training take place?

5) There is a lack of data to prove the effectiveness of the aerobic physical training protocol. It is essential to present the pre and pos Incremental exercise test values.

6) The initial moment at which the echocardiographic evaluation was performed was not mentioned.

7) Why has RV systolic function (estimated by RV fractional shortening) not been evaluated?

8) In line 159, page 7 and line 240, page 11 the statement “measure the cross-sectional areas of the cardiomyocytes…” is not correct. Cross-sectional area and diameter are different assessments.

9) There is a lack of information on the magnitude of increase used to assess the diameter of cardiomyocytes. Authors must present images representative of this assessment. Especially because the values presented are different from those observed in the studies. Sometimes, variability in these analyzes occurs because the heart is not stopped in diastole.

10) The evaluation of the collagen fraction by picrosirius red staining can be more complete if analyzed using polarized light under dark field optics to detect birefringence of collagen fibers. It would not only evaluate the fraction of collagen, but also the composition of its type.

11) A table with the sequence of each prime drawn for qPCR analyzes is required.

12) Why were all data expressed using box plot graphs?

13) Page 14, line 300. “In our study, the phase of cardiac dysfunction by PAH increased col1a1 gene expression, demonstrating the role of this gene in the worsening of cardiac functionality.” It is necessary more evidence for this conclusion. The results of the echocardiographic evaluation demonstrate more changes in morphology and flow, but do not show cardiac dysfunction (i.e., ejection fraction and fraction shortening). It is interesting to note that physical training, despite leading to an increase in LVDD, did not increase the ejection fraction.

14) It is necessary to better understand the result of gene expression for MyH7. The increased expression of this gene observed in the TPAH group was almost observed in the PAH group. Is this increase dependent on physical training or was it induced by treatment with monocrotaline? Is the statistical significance observed only in the PAH group because the variability of the result is less?

Minor issues:

1) Authors need to standardize the way they write São Paulo;

2) Page 10, line 228. * Statistical difference p <0.05. You need to mention which group the comparison was made with;

Reviewer #2: In the work “Preventive training interferes with mRNA-encoding myosin 7 and collagen I expression during pulmonary arterial hypertension”, Mariano T.B. and coworkers describe that exercise had a positive impact on compensated hypertrophy during pulmonary hypertension, partially by the modulation of the extracellular matrix and myosin gene expression on these animals.

However, unfortunately, there are several major concerns regarding the methods, misinterpretation of results, conclusions drawn from the results, and therefore the quality of this study.

Major points:

1. Data do not clearly support the central hypothesis because there is an upregulation of Collagen 1 gene expression, which is related mainly to adverse progression to heart failure – and this is also reported in their introduction.

2. On the other side, they have found that the expression of Mhy7 gene – which encodes Beta –MHC in the heart, is a positive change in trained animals. However, the levels of its gene expression in sedentary and trained are both higher than their matched controls. In addition, reports are suggesting that the b-MHC switch is not responsible for increased contractility in the adapted RV, and even being an adaptative change, this myosin shift is questionable because b-MHC is less powerful overall.

3. The Material and Method section needs a thorough revision. Moreover, ethics in the euthanasia procedure is questionable. In Brazil, the recommendation of CONCEA is to use, for euthanasia, a quantity of pentobarbital corresponding to 3 times the dose recommended for surgical procedures, which correspond approximately to 150mg/kg. In the current work, the authors used 50mg/kg for euthanasia, and some references should be present to justify their choice.

4. The results are somehow confused by figure legends inserted in the main text, instead of a clear description of data – which changes the format in different paragraphs.

5. The results interpretation is also not clear. For example, the statement that “despite persistent right pressure overload, echocardiography confirmed an increase in cardiac function” is based on pulmonary flow velocity, which was enhanced in both trained groups, and LV diastolic diameter, which is lower in both PAH groups. In the discussion, it was not explained how the increase of Mhy7 (which encodes beta-MHC) and collagen type 1 can be related to this “increased cardiac function”.

6. The discussion is too long and also disconnected from the main results and findings coming from its interpretation. There are many considerations regarding other models, pathways, enzymes, but they were not part of the current data present in this manuscript. I suggest a complete restructuration of this part of the manuscript, focusing on the current data and giving a realistic interpretation of the results obtained by the authors.

7. In the expression of those genes, there are many other factors enrolled, like MicroRNAs, inflammation, oxidative stress, epigenetic changes. Many of them are mentioned in the discussion, but no results related to those pathways were present.

In my opinion, the main idea should be reviewed, once the upregulation of Mhy7 and Col1a1gene expression would be related to increased expression of beta-MHC and collagen type 1, respectively, and both have been related to negative prognosis in the progression of compensated cardiac hypertrophy to heart failure. More consistent discussion and interpretation of these data would give the alignment of all parts of this manuscript.

6. PLOS authors have the option to publish the peer review history of their article (what does this mean?). If published, this will include your full peer review and any attached files.

Reviewer #1: No

Reviewer #2: No

---

## [Author Response · Author response to Decision Letter 0]

31 Jul 2021

Authors’ responses to reviewer comments

Reviewer #1: Mariano et al performed a study in vivo to investigate if aerobic training can mitigate (preventive effect) the gene changes in the compensated ventricular hypertrophy phase in monocrotaline-induced PAH rats. The study proposal is relevant; however, several issues must be clarified in order to improve the manuscript. Following the points that must be reviewed.int

Authors’ answer: Thanks for your suggestions. We performed a careful analysis of the results and added other analyzes, such as the PAAT/PET ratio, the Fulton index, and analysis of other types of collagen fibers. A more consistent discussion and interpretation of these data were carried out to promote the alignment of all parts of the manuscript. The title, results and discussion have been restructured.

Major issues:

1) The experimental design is confusing. A topic is needed explaining the model's time course with an illustration. 

Authors’ answer: Thanks for your suggestions. The experimental design has been added. (Page 5, lines: 103-117).

2) There is a lack of information about the monocrotaline treatment.

Authors’ answer: We agree with the reviewer and added more information. (Page: 7, lines: 145-155). Monocrotaline (MCT) is a pyrrolizidine alkaloid derived from the plant Crotalaria spectabilis; it is used to induce PAH in rats by a single subcutaneous injection. The MCT alkaloid is activated by the reactive pyrrole metabolite dehydromonocrotaline (MCTP) in the liver, a reaction that is highly dependent on cytochrome P-450. When administered to rats, MCT recapitulates many features of human PAH. MCT-induced PAH in rats leads to a significant increase in right ventricle (RV) afterload and pulmonary vascular remodeling, as well as greater RV hypertrophy. MCT rat models are the most widely used in vivo PAH model. The rats reproducibly develop pulmonary hypertension and heart failure approximately 4 weeks after a single MCT administration. Below are articles that justify our arguments.

https://doi.org/10.1002/(sici)1099-0461(1998)12:3<157::aid-jbt4>3.0.co;2-k

https://doi.org/10.3109/10408449209146311

https://doi.org/10.1152/japplphysiol.90884.2008

https://doi.org/10.1016/j.yjmcc.2018.04.003

https://doi.org/10.1007/978-1-4939-8597-5_18

https://doi.org/10.1177/2045894020910976

https://doi.org/10.1186/s12931-015-0178-6

2.1) Why was used the dose of 60 mg/kg? 

Authors’ answer: PH was induced in Wistar rats by a single subcutaneous injection of MCT (60 mg/kg), as this dose has already been used in several studies and has shown efficacy in causing the progressive increase of pulmonary vascular resistance. It induces RV hypertrophy (21 days) and RV failure (28 days). Below are articles that justify our arguments.

https://doi.org/10.1152/japplphysiol.90884.2008

https://doi.org/10.1016/j.yjmcc.2018.04.003

https://doi.org/10.1007/978-1-4939-8597-5_18

https://doi.org/10.1177/2045894020910976

https://doi.org/10.1186/s12931-015-0178-6

http://dx.doi.org/10.1016/j.ijcard.2015.11.066

2.2) Were three weeks of treatment sufficient to induce cardiac disfunction?

Authors’ answer: The monocrotaline-induced pulmonary hypertension experimental model with a dose of 60 mg/Kg leads to rapidly progressive RV hypertrophy (21 days). It enables an enhanced sensitivity of detection of right dysfunction due to the drastic magnitudes of change in a short time frame. We evaluated pulmonary artery flow acceleration time (PAAT) and pulmonary artery ejection time (PET) through echocardiographic assessments. To account for heart rate variability in rats, PAAT was adjusted to PET and presented as PAAT/PET. (Pages: 7,8, lines: 166-172). We observed decreased PAAT/PET in the SPAH group when compared to the S group. This decrease was similar to a previous study. Echocardiographic measures of pulmonary arterial pressure, PAAT, were correlated with invasive pulmonary artery pressure (r = -0.74 and r = 0.75, P<0.001). Observed variabilities of the invasive and non-invasive parameters were low; this non-invasive parameter may be used to replace invasive measurements in detecting successful disease induction and to evaluate PAH severity in a rat model. PAAT is also associated with differing degrees of RV dysfunction in patients with pulmonary hypertension, which parallels the relationship of PAAT with pulmonary vascular resistance and pulmonary arterial compliance. This implies that RV afterload is a major contributor to the observed differences in RV function. Below are articles that justify our arguments.

https://doi.org/10.1007/s10554-010-9596-1

https://doi.org/10.1177/2045894020910976

https://doi.org/10.1186/s12931-015-0178-6.

2.3) How was monocrotaline administered?

Authors’ answer: Monocrotaline was administered intraperitoneally, as described in previous studies. This following information has been added. (Page: 7, line: 149). Below are articles that justify our arguments.

https://doi.org/10.1111/j.1440-1681.2008.04936.x

https://doi.org//10.1139/cjpp-2012-0261

https://doi.org/10.1111/j.1365-2613.2006.00475.x

https://doi.org/10.1111/iep.12166

2.4) What was monocrotaline come from? Which company, code and lot?

Authors’ answer: The monocrotaline came from the Sigma Chemical company in St Louis, MO, USA. This information has been added. (Page: 7, line: 150).

 2.4) Was there an adverse effect such as excessive weight loss? 

Authors’ answer: There were no adverse effects such as weight loss. Excessive weight loss is only checked when the animals are in the cardiac cachexia stage, which occurs when there is heart failure. The final body weight values have been included (Page: 8, line: 184 and page: 13, lines: 282-283). Below are articles that justify our arguments.

https://doi.org/10.1038/s41598-017-07236-2

https://doi.org/10.1152/japplphysiol.90884.2008

2.5) What is the limitation of using this PAH model? Important references about this PAH model were lacking.

Authors’ answer: Thanks for your comments. MCT-induced PAH rat models have helped the scientific community to gain insight into the cardiac remodeling process and its pathophysiology. The advantages of MCT models are technical simplicity, reproducibility, and relatively low cost. However, because the animals present heart failure at 4 weeks, it is a model that does not allow the evaluation of the effects of longer period protocols. This limitation have been included. (Page: 18, lines: 434-436).

3) Physical training was performed before and during the development of PAH. Did physical training interfere with the effect of MCT in inducing PAH or did it actually prevent the development of PAH once the condition is established? Is there any data that shows that physical training does not interfere with the action of MCT?

Authors’ answer: Exercise must have interfered by attenuating the effect of MCT in inducing cardiac alterations, as the animals submitted to training had less cardiac hypertrophy demonstrated by smaller areas of cardiomyocytes, lower Fulton index values, and increased PAAT/PET ratios. The latter is an indicator of low afterload and pulmonary vascular resistance. Previous studies have shown a marked decrease in pulmonary artery tunica media thickness in trained rats injected with MCT. This structural change in exercised rats could contribute to an attenuation of pulmonary vascular resistance and may influence the reduction in RV afterload. Moreover, in trained rats injected with MCT, a decrease in interstitial volume and a rise in intra-myocardial vessels volume when compared with the sedentary MCT-injected rats have been reported in other studies. Our findings and these observations suggest that exercise training could promote beneficial changes in RV remodeling induced by MCT. Below are articles that justify our arguments.

https://doi.org/10.1139/cjpp-2012-0261

https://doi.org/10.1111/j.1440-1681.2008. 04936.x.

4) What phase of the 12-hour light/dark cycle did the physical training take place?

Authors’ answer: The exercise was performed in an inverted cycle, in the dark moment, in accordance with previous studies. This information has been added to the text. (Page: 5, lines: 97 and page: 6, lines: 126).

https://doi.org/10.1007/s00395-015-0514-5

https://doi.org/ 10.1113/EP085552

5) There is a lack of data to prove the effectiveness of the aerobic physical training protocol. It is essential to present the pre and pos Incremental exercise test values. 

Authors’ answer: We analyzed the variation velocity (∆) in lactate threshold after MCT application and after 3 weeks of training. The SPAH group showed lower tolerance to exertion than the S group, and the TPAH group had an increase in speed when compared to the SPAH group. This information has been added. (Page: 7, lines: 143-144 and page: 11, lines: 260-264).

6) The initial moment at which the echocardiographic evaluation was performed was not mentioned.

Authors’ answer: The echocardiographic evaluation was performed 48 hours after the exercise. This information has been added. (Page: 7, lines: 158-159).

7) Why has RV systolic function (estimated by RV fractional shortening) not been evaluated? 

Authors’ answer: RV systolic function was evaluated by the indirect measure of RV afterload (pulmonary artery acceleration time [PAAT] and pulmonary ejection time [PET]), as evaluated in previous studies. To account for heart rate variability, PAAT was adjusted to PET and presented as PAAT/PET. PAAT and PET are noninvasive measures of RV afterload that provide accurate estimates of invasive pulmonary vascular resistance, pulmonary arterial pressure, and pulmonary arterial compliance in children with PH and in PH mouse models. In previous studies, it has been established that the PAAT/PET ratio has high sensitivity, specificity, and correlation with invasive measures when compared to the gold standard for PAH assessment. PAAT is also associated with differing degrees of RV dysfunction in patients with pulmonary hypertension, which parallels the relationship of PAAT with pulmonary vascular resistance and pulmonary arterial compliance. This implies that RV afterload is a major contributor to the observed differences in RV function. Intuitively, the development of significantly decreased RV function might lead to a slower rise in ejection velocities early in systole and a longer PAAT. Therefore, the decrease in PAAT/PET in the SPAH group demonstrates functional worsening of the RV. Below are articles that justify our arguments. (Page: 8, lines: 167-172 and page: 12, lines: 269-271).

https://doi.org/10.1007/s11010-016-2937-1

https://doi.org/10.1007/s10554-010-9596-1

https://doi.org/10.1177/2045894020910976

http://dx.doi.org/10.1016/j.lfs.2014.05.008

https://doi.org/10.1016/j.echo.2016.08.013

https://doi.org/10.1152/ajpheart.00336.2006

https://doi.org/10.1086/681267

https://doi.org/10.1152/ajpheart.00979.2001

8) In line 159, page 7 and line 240, page 11 the statement “measure the cross-sectional areas of the cardiomyocytes…” is not correct. Cross-sectional area and diameter are different assessments.

Authors’ answer: This specific assessment was of the area and not the diameter. We corrected the text and presented a figure that demonstrates how this measurement was performed. (Page: 9, lines: 199-203 and page: 13, lines: 292-295)

9) There is a lack of information on the magnitude of increase used to assess the diameter/area of cardiomyocytes. Authors must present images representative of this assessment. Especially because the values presented are different from those observed in the studies. Sometimes, variability in these analyzes occurs because the heart is not stopped in diastole. 

Authors’ answer: Representatives images were added (Figure 4, page: 14 line: 302). In each section, different fields were captured and analysed by choosing the site that exhibited the highest number of cells on a cross‑section. For each rat analyzed, 50 cells were measured. The myocytes selected were cross-sectioned, had a round shape and visible central nucleus, and were located in the subendocardial layer of the RV muscular wall. This was performed with the aim of standardizing the set of myocytes of the different groups. The mean from the cardiomyocyte cross-sectional areas obtained from each group was used as an indicator of cell size. All images were captured by video camera at 40X objective with 400x magnification. This information has been added (Page: 9, lines: 199-203). To complete the hypertrophy assessment, we measured and assessed the Fulton index. The Fulton index is the most common metric used to assess RV hypertrophy. (Page: 8, lines: 186-188).

10) The evaluation of the collagen fraction by picrosirius red staining can be more complete if analyzed using polarized light under dark field optics to detect birefringence of collagen fibers. It would not only evaluate the fraction of collagen, but also the composition of its type. 

Authors’ answer: Thanks for your comments. We performed this analysis; the results were important, as they were in agreement with the analysis of gene expression. (Page: 9, lines: 212-215 and page: 13, lines: 296-301).

11) A table with the sequence of each prime drawn for qPCR analyzes is required. 

 Authors’ answer: No primers were used. We use a customized assay containing Taqman (Applied Biosystems, Foster City, CA, USA) probes specific to each gene. We apologize for the misunderstanding. The number of each assay with Taqman has been added. (Page: 10, lines: 232-241, 248).

12) Why were all data expressed using box plot graphs? 

Authors’ answer: The data that are presented in the box plot were those that had a non-parametric distribution. Statistical tests have been placed in the figure captions. (Page: 28, lines: 655, 670, Page: 29, lines: 685, 692).

13) Page 14, line 300. “In our study, the phase of cardiac dysfunction by PAH increased col1a1 gene expression, demonstrating the role of this gene in the worsening of cardiac functionality.” It is necessary more evidence for this conclusion. The results of the echocardiographic evaluation demonstrate more changes in morphology and flow, but do not show cardiac dysfunction (i.e., ejection fraction and fraction shortening). It is interesting to note that physical training, despite leading to an increase in LVDD, did not increase the ejection fraction.

Authors’ answer: We used the PAAT/PET and the discussion and discussion have been restructured. (Page: 17, lines: 398-427 and page: 18, lines: 428-433). The pulmonary artery flow acceleration time (PAAT) was correlated with invasive n pulmonary artery pressure (r = -0.74 and r = 0.75, P<0.001). This non-invasive parameter may be used to replace invasive measurements in detecting successful disease induction and evaluation of PAH severity in a rat model.

https://doi.org/10.1007/s10554-010-9596-1

https://doi.org/10.1007/s11010-016-2937-1

https://doi.org/10.1177/2045894020910976

http://dx.doi.org/10.1016/j.lfs.2014.05.008

https://doi.org/10.1016/j.echo.2016.08.013

https://doi.org/10.1152/ajpheart.00336.2006

https://doi.org/10.1086/681267

https://doi.org/10.1152/ajpheart.00979.2001

 Physical training increases venous return characterized by an increase in left ventricular pre-load and consequent cardiac remodelling with LVDD increase. The enhanced cardiac contractile function promoted by exercise training is also related to the modest increase in LV muscle mass, LV adaptive cardiac hypertrophy, angiogenesis, and changes in fibrillar collagen content and organization that enhance cardiac pump function. Although the diameter increased, it did not significantly increase the ejection fraction. Such results may be related to the exercise intensity or the reasons why improvements in the LV contractile aspects of hearts in normal rats were insufficient. (Page: 15, lines: 354-359 and page: 16, lines: 360-362).

https://doi.org/10.1007/s00395-014-0454-5

https://doi.org/10.1186/1472-6793-12-11

PMID: 9800883

PMID: 10791020 

14) It is necessary to better understand the result of gene expression for MyH7. The increased expression of this gene observed in the TPAH group was almost observed in the PAH group. Is this increase dependent on physical training or was it induced by treatment with monocrotaline? Is the statistical significance observed only in the PAH group because the variability of the result is less?

Authors’ answer: There was a variability in responsiveness (dispersion of the expression levels as shown by the width of bars in Figure 6) to the effects of monocrotaline in this gene; when evaluated, the p value between S and SPAH groups was 0.0939. Thus, we cannot say that exercise was unable to reverse the effect of monocrotaline in this Myh7 gene. (Page: 14, lines: 304-307 and page: 16, lines: 382-388).

Minor issues: 

1) Authors need to standardize the way they write São Paulo;

Authors’ answer: “São Paulo” has been standardized. (Page: 1, lines: 9,17)

2) Page 10, line 228. * Statistical difference p <0.05. You need to mention which group the comparison was made with; 

Authors’ answer: We apologize for the error. The comparison group has been mentioned. (Page: 13, line: 278).

Reviewer #2: In the work “Preventive training interferes with mRNA-encoding myosin 7 and collagen I expression during pulmonary arterial hypertension”, Mariano T.B. and coworkers describe that exercise had a positive impact on compensated hypertrophy during pulmonary hypertension, partially by the modulation of the extracellular matrix and myosin gene expression on these animals.

However, unfortunately, there are several major concerns regarding the methods, misinterpretation of results, conclusions drawn from the results, and therefore the quality of this study.

Authors’ answer: We performed a careful analysis of the results and added other analyzes, such as the PAAT/PET ratio, the Fulton index, and analysis of other types of collagen fibers. A more consistent discussion and interpretation of these data were carried out to promote the alignment of all parts of the manuscript. The title, results and discussion have been restructured.

Major points:

1. Data do not clearly support the central hypothesis because there is an upregulation of Collagen 1 gene expression, which is related mainly to adverse progression to heart failure – and this is also reported in their introduction.

Authors’ answer: Thanks for your comments. The purpose of this investigation was to evaluate the effects of exercise, in a period that started before the installation of PAH and continued until the RV hypertrophy phase, on molecular mechanisms involving gene expression of myosins and the extracellular matrix. Our findings showed that col1a1 and myh7 genes were altered in the hypertrophy phase in PAH. Despite that preventive training resulted in greater functional capacity, less cardiac hypertrophy, and attenuation of PAH progression, exercise did not modify any of the analyzed genes. The title, results and discussion have been restructured.

2. On the other side, they have found that the expression of Mhy7 gene – which encodes Beta –MHC in the heart, is a positive change in trained animals. However, the levels of its gene expression in sedentary and trained are both higher than their matched controls. In addition, reports are suggesting that the b-MHC switch is not responsible for increased contractility in the adapted RV, and even being an adaptative change, this myosin shift is questionable because b-MHC is less powerful overall.

Authors’ answer: Thanks for your comments. We agree with the reviewer. We evaluated the results of mhy7 gene expression; the high variability of the data showed that the mhy7 gene expression tended to increase in the SPAH group when compared to the S group (p= 0.0939). This suggests that exercise did not change these results. But if we consider the biological effect and the interpretation of the results of the THAP rats, we really cannot say that the tendency of the mhy7 gene expression to increase was a beneficial effect of the exercise. The discussion and interpretation of this data have been reformulated. (Page: 14, lines: 304-307 and page: 16, lines: 382-388).

3. The Material and Method section needs a thorough revision. Moreover, ethics in the euthanasia procedure is questionable. In Brazil, the recommendation of CONCEA is to use, for euthanasia, a quantity of pentobarbital corresponding to 3 times the dose recommended for surgical procedures, which correspond approximately to 150mg/kg. In the current work, the authors used 50mg/kg for euthanasia, and some references should be present to justify their choice.

Authors’ answer: We followed CONCEA's recommendation and used 150mg/kg. It was a typing error. We apologize for the error. The correction and recommendation of CONCEA are now expressed in the text. (Page: 8, lines: 176-177).

4. The results are somehow confused by figure legends inserted in the main text, instead of a clear description of data – which changes the format in different paragraphs.

Authors’ answer: Thanks for the comments. Corrections have been made to the text and figure legends. Legend figures are placed at the end of the manuscript (Pages: 28 and 29).

5. The results interpretation is also not clear. For example, the statement that “despite persistent right pressure overload, echocardiography confirmed an increase in cardiac function” is based on pulmonary flow velocity, which was enhanced in both trained groups, and LV diastolic diameter, which is lower in both PAH groups. In the discussion, it was not explained how the increase of Mhy7 (which encodes beta-MHC) and collagen type 1 can be related to this “increased cardiac function”.

Authors’ answer: Thanks for the comments. We used the PAAT/PET ratio to replace the pulmonary velocity acceleration time (PVAT), pulmonary artery ejection time (PET), and peak flow velocity of the pulmonary artery (PVF). (Page: 8, lines 167-172 and page: 15, lines 336-352). To account for heart rate variability, PAAT was adjusted to PET and presented as PAAT/PET. PAAT and PAAT/PET offer an effective tool in animal models to monitor the establishment and progression of PH and respond to therapeutic interventions. PAAT and PET are noninvasive measures of RV afterload that provide accurate estimates of invasive pulmonary vascular resistance, pulmonary arterial pressure, and pulmonary arterial compliance in children with PH and in PH mouse and rat models. Previous studies have established that the PAAT/PET ratio compares to the gold standard for the assessment of PAH and has high sensitivity, specificity, and correlation with invasive measures. PAAT is also associated with differing degrees of RV dysfunction in patients with pulmonary hypertension, which parallels the relationship of PAAT with pulmonary vascular resistance and pulmonary arterial compliance. This implies that RV afterload is a major contributor to the observed differences in RV function. The development of significantly decreased RV function might lead to a slower rise in ejection velocities early in systole and a longer PAAT. Therefore, the decreases in PAAT/PET in the SPAH group demonstrates worsening of the RV, and physical exercise was able to normalize this parameter in the TPAH group. Below are articles that justify our arguments.

https://doi.org/10.1007/s10554-010-9596-1

https://doi.org/ 10.1177/2045894020910976

http://dx.doi.org/10.1016/j.lfs.2014.05.008

https://doi.org/10.1016/j.echo.2016.08.013.

https://doi.org/10.1152/ajpheart.00336.2006

 Based on the reviewer's comments, the results were re-analyzed, and the discussion has been changed. Mhy7 (which encodes beta-MHC) and collagen type 1 were not altered by preventive exercise (Page: 14, lines: 304-310; page 16, lines 379-393; page 17 lines: 394-427; page 18 lines: 428-433).

6. The discussion is too long and also disconnected from the main results and findings coming from its interpretation. There are many considerations regarding other models, pathways, enzymes, but they were not part of the current data present in this manuscript. I suggest a complete restructuration of this part of the manuscript, focusing on the current data and giving a realistic interpretation of the results obtained by the authors.

Authors’ answer: Thanks for the comments. We agreed with the reviewer, and the discussion has been restructured. (Pages: 14-18)

7. In the expression of those genes, there are many other factors enrolled, like MicroRNAs, inflammation, oxidative stress, epigenetic changes. Many of them are mentioned in the discussion, but no results related to those pathways were present.

Authors’ answer: The interpretation and discussion of the results were reviewed; a more consistent discussion has been presented. (Pages: 14-18

---

## [Decision Letter · Decision Letter 1]

18 Aug 2021

Preventive training does not interfere with mRNA-encoding myosin and collagen expression during pulmonary arterial hypertension.

PONE-D-20-38767R1

Dear Dr. Pacagnelli,

We’re pleased to inform you that your manuscript has been judged scientifically suitable for publication and will be formally accepted for publication once it meets all outstanding technical requirements.

Kind regards,

Michael Bader

Academic Editor

PLOS ONE

Additional Editor Comments (optional):

Reviewers' comments:

Reviewer's Responses to Questions

**Comments to the Author**

1. If the authors have adequately addressed your comments raised in a previous round of review and you feel that this manuscript is now acceptable for publication, you may indicate that here to bypass the “Comments to the Author” section, enter your conflict of interest statement in the “Confidential to Editor” section, and submit your "Accept" recommendation.

Reviewer #2: All comments have been addressed

2. Is the manuscript technically sound, and do the data support the conclusions?

Reviewer #2: Yes

3. Has the statistical analysis been performed appropriately and rigorously? 

Reviewer #2: Yes

4. Have the authors made all data underlying the findings in their manuscript fully available?

Reviewer #2: Yes

5. Is the manuscript presented in an intelligible fashion and written in standard English?

Reviewer #2: Yes

6. Review Comments to the Author

Reviewer #2: Considering the answers and review performed by the authors, I recommend it for publication in this journal.

7. PLOS authors have the option to publish the peer review history of their article (what does this mean?). If published, this will include your full peer review and any attached files.

Reviewer #2: No

---

## [Editor Report · Acceptance letter]

27 Aug 2021

PONE-D-20-38767R1 

Preventive training does not interfere with mRNA-encoding myosin and collagen expression during pulmonary arterial hypertension 

Dear Dr. Pacagnelli:

I'm pleased to inform you that your manuscript has been deemed suitable for publication in PLOS ONE. Congratulations! Your manuscript is now with our production department. 

Kind regards, 

on behalf of

Prof. Michael Bader 

Academic Editor

PLOS ONE